# Landscape Homogeneity May Drive the Distribution of Koala Vehicle Collisions on a Major Highway in the Clarke-Connors Range in Central Queensland, Australia

**DOI:** 10.3390/ani14192902

**Published:** 2024-10-08

**Authors:** Rolf Schlagloth, Flavia Santamaria, Michael Harte, Marie R. Keatley, Charley Geddes, Douglas H. Kerlin

**Affiliations:** 1Koala Research-CQ, Australia Central Queensland University & Koala History and Sustainability Research Cluster, Bruce Hwy, North Rockhampton, QLD 4702, Australia; f.santamaria@cqu.edu.au; 2School of Veterinary Science Gatton, The University of Queensland, Gatton, QLD 4343, Australia; 3Utah Department of Natural Resources, Division of Oil, Gas, and Mining, 1594 West North Temple Street, Salt Lake City, UT 84116, USA; mharte@utah.gov; 4School of Science, RMIT University, Melbourne, VIC 3000, Australia; marie.keatley@rmit.edu.au; 5School of Agriculture, Food and Ecosystem Sciences, University of Melbourne, Creswick, VIC 3363, Australia; 6Koala Research-CQ, School of Health, Medical and Applied Sciences, Central Queensland University, Bruce Hwy, North Rockhampton, QLD 4702, Australia; charley.geddes@cqumail.com; 7Centre for Planetary Health and Food Security, Griffith University, Brisbane, QLD 4111, Australia; d.kerlin@griffith.edu.au

**Keywords:** wildlife vehicle collisions, road strikes, hotspots, habitat fragmentation, koala, *Phascolarctos cinereus*

## Abstract

**Simple Summary:**

After losing their habitat or having it broken up by infrastructure developments, one of the major threats to wild koalas is being hit by cars. Researchers analysed data for 345 koalas that were hit by cars on a 51 km section of the Peak Downs Highway in Queensland, Australia, from October 2014 to November 2023. We found that the spatial distribution of these accidents fit a random pattern along this stretch of road. Two factors seemed to predict where koalas might get hit: the amount of high-quality koala habitat nearby as defined by the local koalas’ tree species preference, and driver visibility along the road. Because the landscape is relatively uniform in terms of landuse and vegetation, koalas can, and do, cross anywhere. More research, including detailed habitat mapping using geographic information systems and ground-truthing, is needed to help protect this important koala population and reduce the number of koala vehicle collisions and associated injury and mortality.

**Abstract:**

After the loss and fragmentation of habitat, vehicle collisions are one of the main threats to the long-term survival of wild koalas. Koala road strike data were analysed for a section of the Peak Downs Highway between Nebo and Spencer’s Gap, west of Mackay, Queensland, Australia. The analysis was carried out on 345 records (October 2014 to November 2023), and results suggested the spatial distribution of koala road strike followed a random pattern along this section of the highway, assuming a Poisson point pattern on a linear network. An analysis of the candidate predictors of koala vehicle collisions, including habitat and road variables, found that the amount of high-quality koala habitat (as defined by the local koalas’ tree species preference) present and the driver visibility were the only significant predictors. The relative homogeneity of landuse and vegetation across this landscape may mean that koalas do not concentrate at specific crossing points. More research, including detailed habitat mapping, is needed into this population, which currently lacks government and conservation attention, to inform mitigation efforts and reduce mortality rates for this potentially nationally significant population.

## 1. Introduction

The impact of vehicle collisions with animals has many facets: death or injury of animals, the resulting ecological consequences, especially, but not only, in the case of protected species [1,2] and the financial cost of the accident to the insurance company and/or the owner of the vehicle. The existence of road blackspots for car accidents, causing human injuries or fatalities, is well established [3,4]. The impacts of accidents have led to significant development in methods to identify areas where accidents occur more often [4,5,6,7].

The Australian Government uses the criterion of 0.13 casualty crashes per kilometre per year over 5 years (a crash where at least one fatality, serious injury or minor injury occurs) to determine the funding eligibility for remedial works on regional and rural roads to address blackspots where high numbers of car crashes occur [8]. However, while it is relatively simple to obtain and analyse data for casualty crashes involving humans and vehicles, it is more difficult to obtain the same data when dealing with animal casualties.

Human casualties are recorded by emergency services and insurance companies. Conversely, animal casualties are only reported if the collision caused significant damage to the vehicle, if the vehicle involved in the collision is insured and is, indeed, reported to an insurance company; even then, the details are often only imprecise and unreliable [9,10]. The species of animal involved may not be known or accurately reported, and the individual animal may depart the scene of the accident injured and either not survive or die later elsewhere, making detailed analysis difficult [11]. However, some animals may reach a wildlife hospital.

Despite been rescued, a study, on koalas taken to a wildlife hospital after surviving an accident showed that these animals were affected by high levels of stress due to the accident, transport and handling [12]. Stress, which is defined as ‘real or interpreted threat to the physiological or psychological integrity of an individual…’ [13] (p. 108) is known to negatively impact animals’ healing processes [14].

The identification of blackspots (in our study referred to as hotspots) for wildlife vehicle collisions (WVCs)—locations along the road with a higher-than-expected aggregation of wildlife vehicles strikes [15]—has been used for the management of wildlife on roads [16]. Species-specific management strategies can be developed and implemented at known hotspot locations [17,18]. For example, studies have shown that a greater number of WVCs occur in areas where there is suitable habitat for a species adjacent to the road corridor [19,20], and ref. [21] found that diurnal species of amphibians were more likely to be involved in WVCs due to the higher diurnal traffic intensity when compared to nocturnal species that crossed the road when the traffic intensity was lower. They also found that slow moving animals were more likely to be hit than faster moving species, while others [22] suggested that daylight saving time and associated changes to the timing of commuter traffic has the potential to reduce collisions with wildlife, especially nocturnal animals. However, other studies have shown that the impact of traffic intensity on WVCs is less than the impact of increasing road network densities [23], especially for more mobile species.

Hotspot identification can also allow for predictive modelling, with habitat-use metrics and forage availability and protective foliage cover measures found to be indicative of where animals are more likely to be involved in WVCs [18,24]. Unfortunately, determining hotspots can be difficult due to a paucity of data about WVCs (particularly spatial data), due to a lack of reporting for the majority of species. However, certain high profile species, such as the koala (*Phascolarctos cinereus*) in Australia, are associated with high reporting rates in care facilities or Government records [25].

Koalas are an endangered species in Australia, with vehicle strike a key threatening process [26,27,28]. Koala populations are not only affected directly by WVCs, but also by the fragmentation effect that roads have on habitat [29]. Roads can have a significant barrier effect for wildlife, resulting in increased fragmentation of both habitat and populations [30]. Habitat fragmentation forces koalas to travel more frequently to sustain themselves [31], therefore increasing the risks of vehicle strike or dog attack while on the ground [19,23].

Furthermore, the construction or widening of roads has a direct impact on koalas and other wildlife through loss of habitat as well as an indirect impact on their home ranges. The affected animals may need to increase home range sizes or make other adaptations which can expose them to the threat of WVCs [32]. Importantly, the structure of the landscape and road design plays an important part in this threatening process, and they need to be incorporated into koala conservation [32]. Another study [19] showed that WVCs involving male koalas were more frequent than those involving females, mainly due to the increased movement of males during the breeding season, and more generally due to the larger area of male home ranges. Koalas residing in urbanised landscapes often traverse roads to access fragmented habitat patches [18], increasing their susceptibility to WVCs [23].

As part of the realignment of the Peak Downs Highway (PDH) at Eton Range [33], a study of the koalas inhabiting Central Queensland’s Clark-Connors Range was undertaken with the goal of better understanding future management options around the highway, and in the greater region. These populations have only received limited attention in the past [34], but a recent study [35] has called for urgent action to address the issue of koala WVCs in this area. The area is relatively remote, with low human population densities. Gathering WVC data requires a continual monitoring effort, and this has only been possible in this area due to the work of a small number of dedicated citizen scientists and conservationists.

While prior studies used either limited data or conceptual models to identify a number of significant koala WVC hotspots along the PDH [36,37,38], this study used an updated dataset (345 koalas). Here, we have also utilised specific statistical tools to reassess the distribution of koala WVCs along the PDH. The objectives of the study were to undertake modelling to predict koala vehicle collision hotspots on the Eton to Nebo stretch of the PDH and to provide important information which will be beneficial to the conservation of local koala populations in the wider Clarke-Connors Range region. We hypothesised that koala WVCs would cluster at points of significant habitat values, high water availability and/or locations with high vehicular speeds and limited driver visibility. The identification and prediction of hotspot occurrences can inform management and planning to avoid and mitigate against WVCs with positive animal welfare, conservation and human safety outcomes.

## 2. Materials and Methods

The study site was a 51 km section of the PDH, over the Clarke-Connors Range between Spencer’s Gap (approximately 10 km SSW of Eton) and Nebo, west of Mackay in Central Queensland (Figure 1). The PDH is the main service road supplying the mining industry of the Bowen Basin and agricultural activities in the region; Nebo is a significant centre for mining, with eight mines currently operating in the area [39]. As the highway travels through the study site, it is effectively a single high-speed (100 km/h) road. Traffic data for the PDH are limited; however, volumes extrapolated from survey data, provided by TMR CQ-region, show a continual increase in the average annual daily traffic reflected by approximately 4162 vehicles/day in 2020, 4306 (2021) and 4460 (2022) with a 3.70 percent growth in the last five years. Apart from some limited and short sections of wildlife-exclusion fencing associated with several bridges [35], nearly all other fencing is of a kind associated with livestock farming and usually consists of multiple strains of barbed wire. There is one state-controlled road (Blue Mountain Road) intersecting with the highway approximately 15 km from Eton, and another (Suttor Development Road) intersecting approximately 6 km from Nebo. All other roads intersecting with the highway along this section (~11) are minor roads terminating after less than 2 km.

The study site traverses 26 different Regional Ecosystems [40], 20 of which, to various degrees, are likely to contain koala habitat tree species. Regional Ecosystems (REs) are a standardised biodiversity classification system developed by the Queensland State Government, defined as “vegetation communities in a bioregion that are consistently associated with a particular combination of geology, landform and soil” [41]. The main koala fodder tree species likely to be found in the area are *Eucalyptus tereticornis*, *E. camaldulensis*, *E. coolabah*, *E. melanophloia*, *E. crebra*, *E. populnea*, *E. drepanophylla* and *E. platyphylla* [42].

Records of koala road strikes along the study site were obtained from various sources, primarily wildlife carers (22) and the Queensland Department of Transport and Main Roads (TMR). Two individuals were responsible for 82% of koalas recorded: one was the operator of the wildlife care facility covering the study site, the other a dedicated citizen scientist living in the local area. All records were extracted from BioCollect, project: Koala Mapping Mackay, Whitsundays and Central Queensland Areas; Padaminka Nature Refuge Koala Rescue and Care (https://biocollect.ala.org.au/project/index/760dc5ab-163a-4779-9e09-bca599c9847b—accessed on 25 November 2023). The recording of WVC data by two individuals only may introduce the possibility of some bias; however, we note that without any of these records we would have no data for this population and no awareness of the significance of koala WVCs on this road.

### 2.1. Distribution of Koala Road Strikes

The distribution of koala road strikes along the Nebo–Spencer’s Gap stretch of the PDH was examined to identify potential WVC blackspots. Initially, we made use of KDE+ [43], an application of kernel density estimation for point patterns on a linear network, which incorporates bootstrap sampling to identify statistically significant clustering. To verify the results of this analysis, the locations of koala road strikes were converted to a point pattern on a linear network using the linnet.spatstat package version 3.0-3 [44] in R version 4.2.1 [45]. We estimated the linear K function [46] for koala road strikes and tested the significance of results against an envelope of 1000 simulation results assuming a homogeneous Poisson process [47].

### 2.2. Predictors of Koala Vehicle Strike Locations

We were constrained by limitations in driver visibility data, as these were only available for a limited series of chainage points (points at a fixed interval) along the highway. As a result, analysis to identify predictors of koala vehicle strike was conducted across chainage points located every 5 km along the road resulting in 11 chainage points (the analysis was also run using other distances between chainage points, and therefore other numbers of chainage points, with no significant impact on the results; reduced distances between chainage points and smaller buffers were impacted by spatial autocorrelation between points). A buffer (2.5 km) was applied around each chainage point, and koala vehicle strikes were counted within each buffer and apportioned to the nearest chainage point. Co-variates were also produced, including measures of the koala habitat values of areas within the buffer, proximity to watercourses, vehicular speed, road width and driver visibility.

### 2.3. Koala Habitat Values

Koala habitat ranking based on REs, in conjunction with targeted ground-truthing, was undertaken within the study area using an expert-driven qualitative classification (Appendix A). Koala habitat was initially ranked into five koala habitat quality classes (1—poor, through to 5—very high quality). Due to a high degree of collinearity, we combined habitat classes 4 (high) and 5 (very high) into a single ‘High quality’ class for further analysis. The lower three habitat classes were also combined: 1 (poor), 2 (low) and 3 (medium), were combined to form a single ‘Low quality’ class; however, areas of high-quality koala habitat tend to lack low-quality koala habitat and vice versa. As a result, the low-quality class was determined to be significantly inversely correlated (r = −0.99, df = 9, *p* < 0.001) with the high-quality class, so was removed from further analysis, leaving a single ‘high-quality koala habitat’ variable.

We further extended our analysis of koala habitat mapping to examine a number of landscape fragmentation and habitat availability indices using the QGIS LecoS plugin for patch and landscape statistics [48]. However, analysis of the correlation matrix of these measures showed a high degree of collinearity between the metrics, with *r^2^* values typically > 0.8 (or inverse correlation with *r^2^* values typically < −0.8). Therefore, the final analysis included the amount of high-quality koala habitat contained within each 2.5 km buffer only.

### 2.4. Proximity to Watercourses—Distance of Mapped Watercourse Lines

Prior studies have noted a relationship between koala occupancy and rainfall, or distance to water features [49]. In inland parts of their range, koalas have been noted to aggregate around bodies of standing water [50] or make extensive use of artificial drinking stations [51]. However, it has also been observed that in more coastal regions, koalas do not utilise these artificial drinking stations, perhaps suggesting that access to a source of water is only required during periods of water stress [52]. The PDH occupies a more northerly location than the studies cited, so we are uncertain as to the role standing water plays in the distribution of koalas in this region. To account for the possibility that koala road strike may be impacted by proximity to standing water, we included a proxy measure.

Mapping of local watercourses was sourced from the Queensland Government (“Watercourse lines—North East Coast drainage division—central section” © State of Queensland Department of Resources 2021). For each chainage point, we calculated the sum length of all mapped watercourses within each 2.5 km buffer.

### 2.5. Vehicular Speed

For the three variables used to assess the impact of traffic along the highway on koala road-kills, chainage points were mapped every 100 m across the length of the study site. This information was provided by the Queensland Government Department of Transport and Main Roads (TMR). Vehicular speed was defined as the mean legal speed limit (km/h) as the road passed across each 2.5 km buffer. Koala vehicle strike mortality has previously been demonstrated to be highest on roads with high traffic volumes and high speed limits [19].

No actual speed measurements for all chainage points were available. Instead, the assumption was made that most vehicles would travel at or around the legal speed limit, and that breaches of these limits would most likely apply uniformly across all chainage points. This is an obvious simplification made in the absence of a more robust measure. As a highway, the speed limit was generally uniform and high; along most of the highway the limit was 100 km/h, excepting for an 800 m section of road where the limit gradually increased from 60 to 100 km/h as the road passed out of Spencer’s Gap, and a 100 m section at 90 km/h, followed by 700 m at 80 km/h as the road enters the town of Nebo.

### 2.6. Road Width

Wider roads represent a wider barrier to koala movement; they may also allow for higher density traffic flows [53]. The width of the highway varies, with overtaking lanes in alternate directions along multiple sections. As with the speed limit, mean road width was calculated (in metres) for each 2.5 km buffer.

### 2.7. Driver Visibility

We defined driver visibility as the distance a camera positioned to reflect driver visibility could view until it was obstructed by changes in the road (e.g., curvature, elevation). These measurements (expressed in metres) were transcribed using mobile laser scanning HawkEye [54] data and footage from the Digital Video Road program supplied by TMR. The system combines several cameras in a regular passenger vehicle and continually records while the vehicle is travelling along the highway. Cameras allow the gathering of various data in both directions (classified here as ‘Visibility East’ and ‘Visibility West’).

While we acknowledge that koalas are known to mostly move at dusk and dawn [55], and drivers’ ability to see in the distance is likely to be restricted by a lack of daylight, driver visibility was estimated using measures taken during daylight hours, due to the lack of an alternative. However, we assume that relative differences in driver visibility between chainage points during daylight hours would be similar to those observed at night.

There was a reasonable degree of correlation between ‘Visibility East’ and ‘Visibility West’ (*r^2^* = 0.697, *p* < 0.001). However, for some sites there were large differences between directions, and given we had no information as to which direction vehicles are travelling prior to a koala strike, driver visibility was included in modelling using the sum of ‘Visibility East’ and ‘Visibility West’, averaged (mean) across each 2.5 km buffer.

### 2.8. Statistical Analysis

Explanatory variables (summarised in Table 1) were tested for spatial autocorrelation using Monte Carlo simulation of Moran’s I. Explanatory variables were incorporated into the analysis through a Poisson regression model, and a Bayesian spatial filtering model (following [56,57]) with 10,000 iterations to allow for spatial autocorrelation, using the ngspatial package version 1.2-2 in R [58]. For these analyses, we defined our adjacency matrix such that each point along the road was identified as adjacent to the point before and after.

## 3. Results

We analysed records of 345 koalas struck by vehicles on the PDH between 10 October 2014 and 25 November 2023 (Figure 1). There has been a significant growth in the numbers of koalas recorded from 4 in 2014 to 145 in 2023 (through 25 November; Figure 2). Koala detections show evidence of an annual cycle, with increased numbers of koalas in the period from August to December. 

A vehicle strike resulted in the death of the koala in 82.9% and injury in 15.8% of cases. The condition of the koala in the remaining 1.3% of cases was unclear. Two-hundred and fifteen individuals (45.2%) were male and 61 (17.7%) were female, with the sex of 37.1% of individuals unknown (often due to the poor condition of the carcass when the animal was recovered). All koalas that were recovered in a sufficiently good condition to be assessed were reported as generally healthy, with one exception: one koala was reported as having a stained bottom (indicative of possible chlamydial infection [59]).

The mean distance to the nearest-neighbour for road-strike locations was 73.7 m. The KDE+ tool did not detect any hotspots along the road, suggesting minimal significant clustering (note that this method utilises kernel bandwidths < 1 km [60]). Heatmapping using kernel density estimation was similarly suggestive that koala vehicle strikes are evenly distributed along the study site (Figure 3). This result is supported by an estimation of the linear K function for a point pattern on a linear network [46], which showed no significant clustering at distances of less than 6 km, but some evidence of clustering at greater distances (scales of 6 to 15 km; Figure 4).

A further analysis was conducted, aimed at identifying parameters that may contribute to the occurrence of koala road strikes. Tests showed our ‘high-quality koala habitat’ measure was not spatially autocorrelated (*p* = 0.172), nor was the proximity-to-watercourses measure (*p* = 0.282) or the speed limit (*p* = 0.18). However, driver visibility (*p* = 0.022) and road width (*p* = 0.014) were found to be spatially autocorrelated. A generalised linear model with a Poisson error structure using the non-spatially autocorrelated predictors found that the number of koala vehicle strikes was significantly (and positively) associated with the amount of high-quality koala habitat. Other explanatory variables in the model were considered insignificant (however, the speed limit was considered significant if the amount of high-quality koala habitat was excluded from the model). To incorporate the spatially autocorrelated predictors, we made use of a Bayesian spatial filtering model (Table 2).

Chainage points associated with larger areas of high-quality koala habitat mapped in the local landscape were significantly more likely to be associated with koala road strike (Figure 5), as were chainage points with lower driver visibility. Other explanatory variables were not significant.

## 4. Discussion

Koala vehicle collisions are a significant cause of mortality for the endangered koala [61,62]. It is, however, uncertain whether this increase in koala roadkill is reflective of growth in the local koala population, or the increased detection of koalas hit by a vehicle due to increased vigilance of locals in the community.

We observed seasonal dynamics in the occurrence of collisions, with a peak from August to December. This pattern has been observed in other koala populations in South East Queensland (SEQ) [61,62], mid-coast New South Wales [63] and Victoria [64], and is hypothesised to result from increased activity during the breeding season.

Prior studies have demonstrated that within urban areas in SEQ, koala vehicle collisions may occur at certain locations with a greater frequency than expected by chance [65]. In the current study, we expected that koala road strikes along the Nebo–Eton section of the PDH would similarly cluster around locations significant to koalas; we hypothesised these would include areas of preferred koala habitat and increased water availability, as well as high vehicular speeds and low driver visibility. However, we were unable to detect any statistically significant hotspots across our study site. This may reflect a random pattern in the spatial distribution of road strikes (except at very large thresholds), with koalas equally likely to be hit by a vehicle at any location along the highway.

In comparison to the SEQ context, the relative homogeneity of the landscape does not appear to concentrate koalas to particular crossing points in the PDH context. Much of the landscape along the study site constituted large patches of native vegetation in a mosaic with cleared areas for grazing stock (with isolated trees). Other landuses were limited. Conversely, the comparatively heterogeneous landscape of SEQ (an urban and peri-urban setting, similar to that in which many road ecology studies take place), with limited koala habitat areas coexisting within a matrix of housing, industrial parks, commercial centres, infrastructure and open space, means that koalas are concentrated within fragmented patches of habitat in the landscape, and funnelled to cross roads at a more limited set of points across the road network.

Previous studies have demonstrated conceptually similar ideas. A study of prairie-chickens found differences in mortality comparing populations in Oklahoma and New Mexico; the authors considered these differences to be due to a greater level of fragmentation of the landscape in Oklahoma (associated with higher mortality rates), with the large pastures of New Mexico offering a more homogeneous landscape [66]. In a similar vein, differences in vertebrate wildlife mortality were associated with the heterogeneity of the landscape along a highway in Kansas [67]; in that study, mortality was greatest in landscapes with large contiguous tracts of prairie, as compared to areas with a patchwork of smaller stands. Both studies illustrate differences in wildlife mortality rates in response to landscape heterogeneity. We consider our study site to be relatively homogeneous, lacking in features that might concentrate koala road-crossing attempts to particular locations. In fact, our study demonstrates how the distribution of mortality events can also be impacted by a lack of heterogeneity in the landscape. Here, the driver for variation in the distribution of vehicle strikes is the presence or absence of habitat along the road corridor.

The homogeneity of the landscape also made the identification of any predictors of koala vehicle strike locations difficult. Given the uniformly high speed along most of the study site, there is limited variability in predictive power attributable to the speed limit. Studies have shown that a reduction in speed reduces WVCs [10,68]; however, a trial in SE Queensland showed that drivers are unlikely to adhere to enforced reduced speed limits which meant that a comparable number of koalas were killed before and after the speed limit changes. Hence, attempting to influence driver behaviour through speed controls only is unlikely to significantly reduce koala road deaths [23]. It was also shown that a lower actual speed resulted in a slight increase in koala survival rates. In our study, we note that koala vehicle strikes were lowest at the extremes of the study site, areas which have lower speed limits due to the ascent up the Clarke-Connors Range to the northeast, and the entry into the township of Nebo to the southwest. This pattern was apparent in Figure 5, with reduced koala WVCs at the extremes of the study site where the speed limit was lowest. Unfortunately, this reduction in WVCs coincided not only with a reduced speed limit but also with the larger effect of changes in the amount of high-quality koala habitat, which means that it is difficult to determine whether the reduction in deaths was due to a reduction in speed only.

Similarly, our proxy measure for the proximity to watercourses may have lacked sufficient variability to capture the role of watercourses in creating pinch points for koala road exposure and therefore koala vehicle strikes. A physical examination of the watercourses to determine the amount of standing water may improve this measure, as many of the mapped watercourses may be intermittent.

Road width was not a significant predictor of vehicle strikes. This was potentially surprising, as larger road widths can be indicative of overtaking lanes for traffic; with overtaking lanes there may be a propensity towards increases in vehicular speed [69,70]. Additionally, large road widths mean koalas are on the road and exposed to potential strike for a longer duration of time. Increased risks to koalas on wider roads may, however, be offset by an increased driver visibility or reduced traffic volumes as vehicles spread out over an increased number of lanes.

Driver visibility was a significant predictor of koala WVCs. This is a finding that has been demonstrated in prior studies [71]. When the obstacle is a small, grey koala on a dark background, it is unsurprising that more collisions occur in areas where drivers have less time to detect and react to koalas on the road. Additionally, we do not have any knowledge of when koalas were struck; it may be that most collisions occur at night when visibility is further impaired. It is also worthwhile noting limitations due to how the data were supplied. As these were provided as measurements every 100 m along the road, there was the possibility of fine-scaling this analysis. However, the drawback was that the koala habitat data would have to be sampled over much smaller areas—if, for instance, the analysis was scaled back to predict koala vehicle strikes every 100 m along the road, habitat measures would likely have to be taken from a 100 m buffer around each chainage point—which results in a great degree of spatial autocorrelation in the habitat data, plus a much smaller view of habitat around each chainage point. Ultimately, we decided to average the visibility data over a greater distance to allow for a more coherent examination of the role of habitat in koala vehicle strikes.

The aerial extent of high-quality koala habitat was also a significant predictor. Counts of koala vehicle strikes were greater in areas with a large area of high-quality koala habitat. This makes intuitive sense; koalas are more likely to occur in areas with a higher proportion of patches with contiguous preferred habitat [72], noting that collisions were not sufficiently clustered to represent hotspots. The preferred habitat can support high densities of koalas, and with higher densities of koalas, there is a higher probability of vehicle strike [73]. Koalas are more likely to persist in landscapes with greater than 50% high-quality (primary) habitat configured in large patches [30], and [74] found that it was essential to protect remaining core areas of high-quality habitat and scattered habitat patches which provide connectivity and enhance opportunities for safe koala movement between habitat patches intersected by main roads.

However, we recognise that the scale of currently available koala habitat mapping is coarse, and there may be fine-scale changes in habitat that are not currently captured. For example, there may be particular vegetation groups that are not fully demarcated by the mapping that may be of particular significance. Koalas may also make use of singular significant trees in the landscape. In a similar study of koala vehicle strikes in Ballarat in Victoria, the first author was able to map 10,000 individual trees and examine koala usage of individual trees. This detailed data enabled an analysis of habitat usage by individual koalas, on a scale of both individual patches and individual trees. It showed that in the Ballarat context, koalas demonstrate very restricted use of habitat, for example, as few as 2–5 individual trees [71]. Vegetation mapping at this scale is currently not feasible in this study area but should be prioritised; greater clarity on the role of habitat influencing koala vehicle strikes could be achieved with a greater detail of koala habitat mapping.

There is commonly significant bias within historical datasets, which lessens the utility of a time-series analysis. Koala counts are obviously a function of the frequency with which motorists are sufficiently aware and motivated to detect and report koala vehicle strikes as they travel along the road [75]. As mentioned, of the 345 koalas recorded as having been struck within the study site, two individuals were responsible for 283 of the koalas recovered. One of the two individuals mentioned moved to the area in 2013 but did not start to report the koalas she spotted until several years later. On the probability of detecting a koala vehicle strike event, she stated it “would make a difference how many days a month I go into town, and how much time I have to stop and check spots where it is not clear whether it is a koala that has been struck”. Similarly, if one or both of these people were away on holiday, etc., the number of koalas reported would decline. The records analysed do not, therefore, represent a complete inventory of all koala vehicle strikes along this stretch of road. For that reason, we have focused the main finding of this study on the location, not the frequency of koala vehicle strikes.

However, this dataset represents the best information currently available on koalas in this area, and the potential impacts of roads on the population. The impacts of roads on this population warrants further investigation. This region may be home to a significant koala population, but the absence of basic population data has hampered recognition. Research has estimated that vehicle strike is associated with 365 koala arrivals to wildlife care facilities across the entire SEQ region (>20,000 km^2^) each year [62]. Across this study site, a single 51 km stretch of highway, records show that since 2018 an average of 72 koalas have been hit by cars each year, with 145 koalas recorded for 2023 (through 25 November). To date, much of the focus of legislation and management has been towards koalas in the rapidly urbanising SEQ, resulting in populations across the rest of the state being comparatively neglected.

The continued strength of the Australian resources sector, as well as agriculture and the burgeoning alternative energy sector, has meant increased traffic flows on regional highways [76], which has implications for threatened endangered wildlife such as the koala. A renewed focus on regional Queensland from policy makers, in collaboration with partnerships with industry and property owners and managers is needed to increase our knowledge of koala populations outside SEQ. In addition to a consideration of the direct impacts of development (e.g., habitat clearing), development assessments through the Environmental Protection and Biodiversity Conservation Act (1999) also need to consider how increased traffic flows generated by the development and operation of large-scale projects may impact listed threatened species living along adjacent transportation corridors. An increased focus on landscape management, through industry collaboration and Government contributions, is needed to safeguard the species and improve the effectiveness of koala-sensitive infrastructure along the Peak Downs Highway.

## 5. Conclusions

Koala road strikes along the PDH are a serious issue for koala conservation in the local population and deserve much greater attention. This population attrition due to WVCs observed over a period of time is of great concern for this endangered species. This is potentially a koala population of national significance for which we lack basic knowledge of its abundance and health.

A key finding of this work is that the spatial distribution of koala WVCs across the length of the road follows a random pattern. We hypothesise that this is driven by the homogeneity of the surrounding habitat, which means koalas are less likely to be ‘funnelled’ towards particular pinch-points on the road network and therefore may potentially be struck by vehicles anywhere along this road. This likely has wider applicability; a greater diversity of landuses (especially where the landscape includes a mix of both hospitable and inhospitable landuses for the species in question) along transport corridors may funnel all manner of species to particular points on the network that are adjacent to areas of preferred habitat. This may be a key driver of the formation of WVC hotspots. Conversely, in less diverse landscapes, where the surrounding landscape is all effectively hospitable to the species in question, individuals can potentially enter the transport corridor at any point. It is also worth mentioning that while our analysis focused on a road network, the same issues likely apply to rail corridors.

Management to reduce WVCs, in particular separating koalas and vehicles, will be difficult as koalas can potentially cross anywhere. The avoidance and mitigation of WVCs are best performed at the planning phase by trying to avoid areas of high-quality koala habitat, or by constructing or retrofitting protective and/or diversionary infrastructure, which is an approach that has already been started by the managing authority in several locations along the highway [35]. Improvements targeting traffic speed and driver visibility, particularly in the planning phase, would have a significant benefit, as would interventions to improve driver attention and awareness of the potential for koalas to be on the road. A more comprehensive understanding of the local koala populations could also help in planning, as would the input of detailed, field-collected vegetation mapping. Novel technologies including AI-powered wildlife detection and recognition linked to driver and/or wildlife-targeted warning systems (e.g., [77,78]) also promise to provide innovative ways to reduce wildlife vehicle collisions.

## Figures and Tables

**Figure 1 animals-14-02902-f001:**
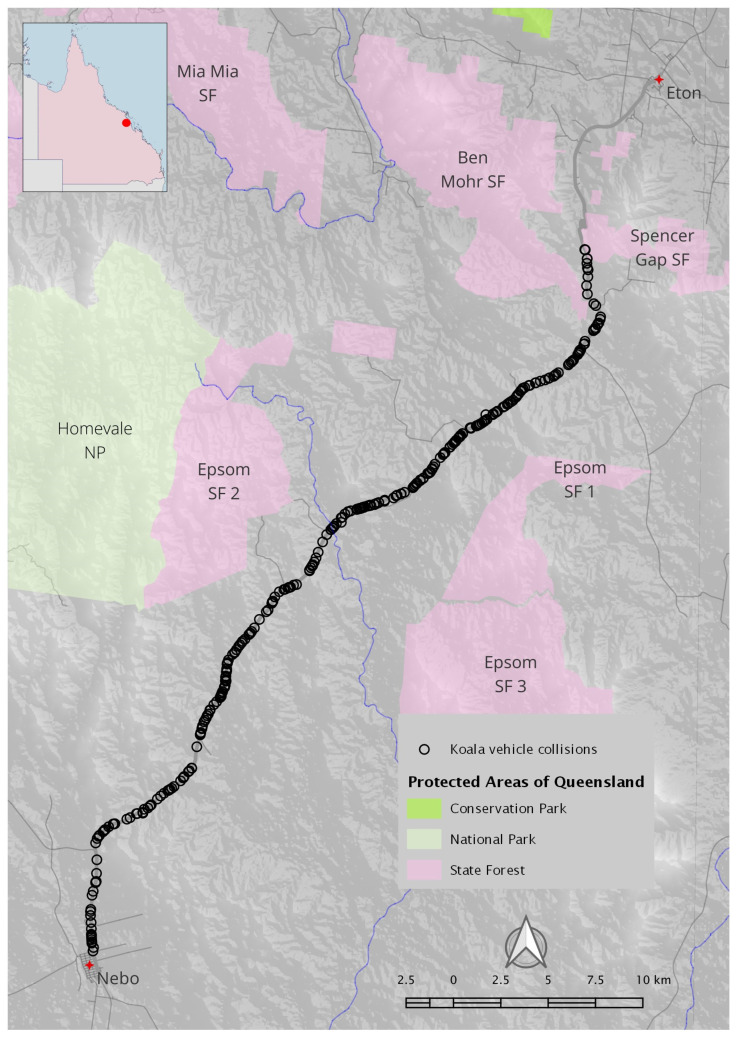
The study site follows the Peak Downs Highway over the Clarke-Connors Range between Spencer’s Gap and Nebo, west of Mackay in Central Queensland. Black circles represent recorded locations of koala vehicle collisions between October 2014 and November 2023. Insert shows study site (red dot) in relation to the state of Queensland.

**Figure 2 animals-14-02902-f002:**
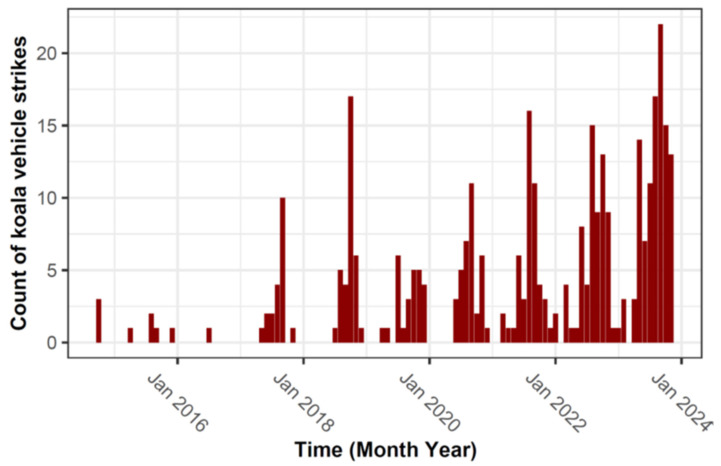
Detections of koala vehicle strikes from October 2014 to November 2023.

**Figure 3 animals-14-02902-f003:**
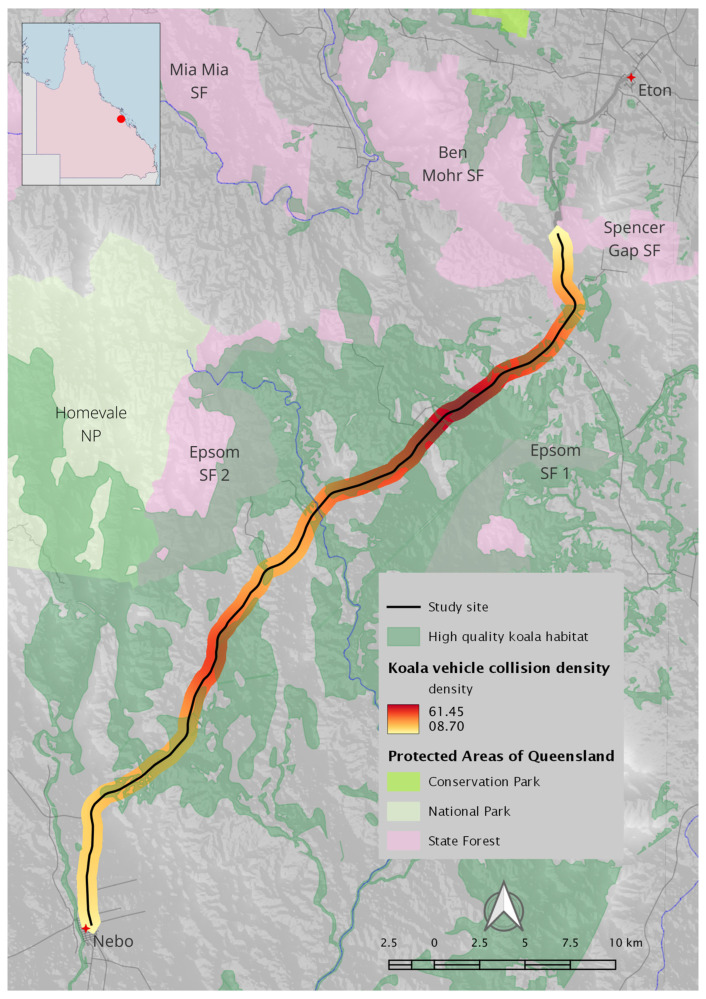
Heatmapping of koala vehicles collisions using kernel density estimation finds high densities of vehicles strike across the study site. Note that identification of hotspots assumed WVC locations were distributed in one dimension (along the study site) rather than in two dimensions. Insert shows study site (red dot) in relation to the state of Queensland.

**Figure 4 animals-14-02902-f004:**
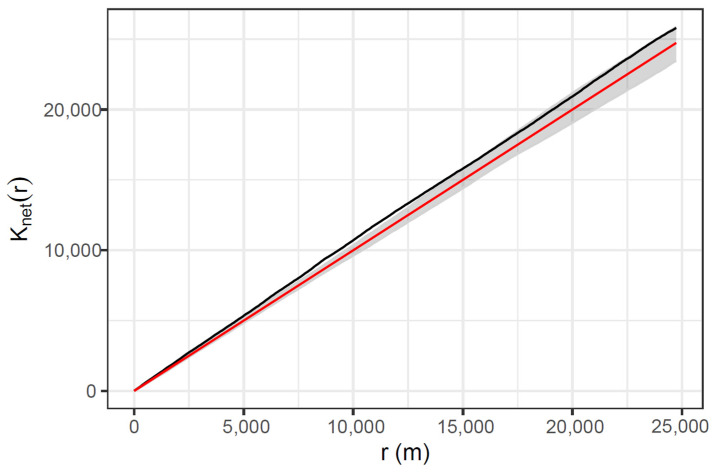
Estimation of the linear K function for a point pattern on a linear network ([46]; black line). The observed linear K function is compared to 1000 simulation results assuming a homogeneous Poisson process (grey shading, with the red line indicating the theoretical expected value for the simulation results). Results suggest a random distribution of vehicle strikes at scales less than 6 km, but some evidence of clustering at greater distance scaling.

**Figure 5 animals-14-02902-f005:**
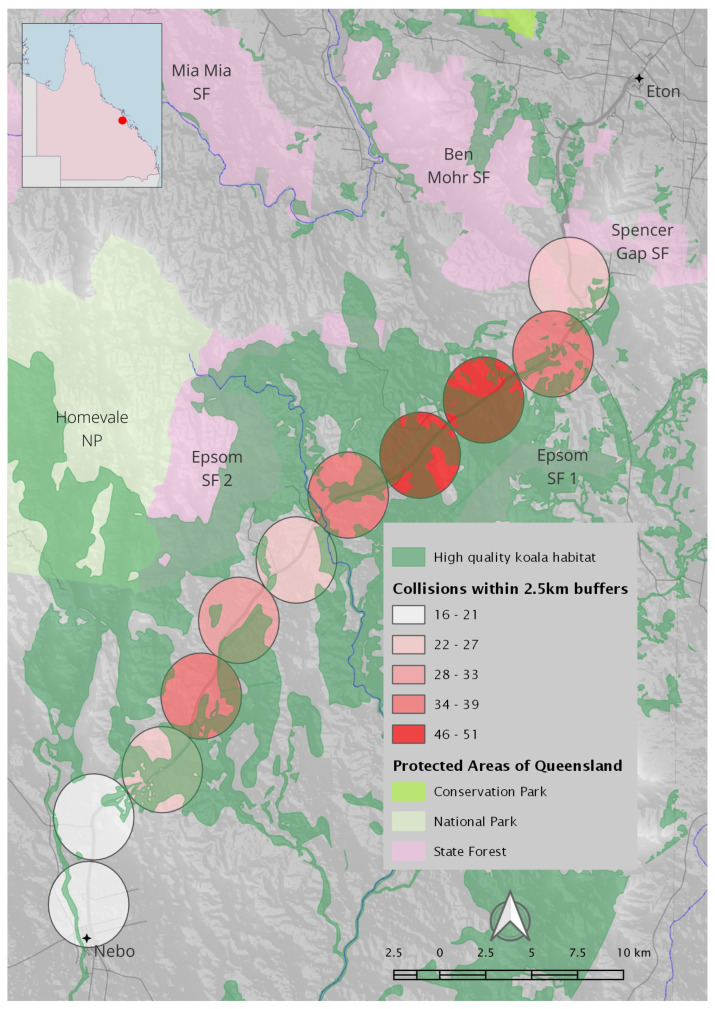
Chainage points utilised for analysis. A 2.5 km buffer was placed around each chainage point, and koala vehicle strikes within each buffer were enumerated. Brown shading indicates mapped areas of high-quality koala habitat values. Koala vehicle strikes were associated with greater areas of high-quality koala habitat; however, these were not sufficiently clustered to represent hotspots. Insert shows study site (red dot) in relation to the state of Queensland.

**Table 1 animals-14-02902-t001:** Summary statistics for explanatory variables used in the Bayesian spatial filtering model.

Variables Included in the Model	Mean	Min	Max
Speed limit (km/h)	97.63	96.15	100
Driver visibility (m)	213.49	158.04	426.92
Road width (m)	11.11	9.02	14.35
Length of mapped watercourses (km)	38.3	20.8	59.6
Area of high-quality koala habitat (ha)	549.4	93.5	1100.4

**Table 2 animals-14-02902-t002:** Bayesian spatial filtering model output. Model parameters were tested to determine the significance of any impact on the likelihood of koala road strike at a given chainage point. Significant parameters were identified where the coefficient estimate was significantly different to zero (i.e., where the lower–upper coefficient estimate did not overlap with zero—identified with *).

	Coefficient Estimate	95th% Confidence Intervals	
		Lower	Upper
Model intercept	27.98	0.12	57.4
Speed limit (km/h)	−0.03	−0.06	0.002
Driver visibility (m)	−0.007	−0.013	0.002
Road width (m)	−0.014	−0.13	0.1
Length of mapped watercourses (km) *	−0.03	−0.063	0.002
Area of high-quality koala habitat (ha) *	0.0007	0.0002	0.0012

## Data Availability

Koala roadkill data are available at: https://biocollect.ala.org.au/project/index/760dc5ab-163a-4779-9e09-bca599c9847b—accessed 25 November 2023.

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
