# Peer review of "Landscape Homogeneity May Drive the Distribution of Koala Vehicle Collisions on a Major Highway in the Clarke-Connors Range in Central Queensland, Australia"

_animals, 2024, doi:10.3390/ani14192902_

Round 1

Reviewer 1 Report

Comments and Suggestions for Authors

There are already numerous studies on the spatial distribution of roadkills, so this research area is not new. I found few innovative aspects in this study and have suggestions for the authors to enhance the manuscript.

Introduction.

Page 3, Line 110-116. The authors claim to have utilized an updated dataset and advanced statistical tools in this study. However, I disagree. They recorded only 345 koalas between October 10, 2014, and November 25, 2023, and included just five general factors in the model: koala habitat values, proximity to watercourses, vehicular speed, road width, and driver visibility.

Materials and Methods.

Please provide more detailed information about the road, including whether it is fenced or unfenced and the traffic volume. This information is crucial for understanding the impact of road traffic on roadkill. Regarding these five factors, I suggest creating a table to list the maximum, minimum, and average values for each factor.

1.Page 6, Line 240, please remove 3. Results

Results

1.Please relocate much of the content to the "Discussion" section, as it seems you've conflated results and discussion. Additionally, avoid citing references here, such as those on Page 6, lines 245-251.

Discussion

1.Page 13, Line 388-389. You mentioned, However, we recognize that the scale of currently available koala habitat mapping is coarse, and there may be fine-scale changes in habitat that are not currently captured.I believe this is a potential direction for your future research, so please include it in the conclusion.

 Conclusion

1. I believe it appears too lengthy; consider condensing it into one paragraph.  

2. Most of the second paragraph seems to fit under "Discussion"; please include future research directions at the end of the "Conclusion."

Correction: Page 11, Figure 2 should be Figure 5.

Reviewer 2 Report

Comments and Suggestions for Authors

++Title:

The phrase "may drive" suggests uncertainty, which is acceptable in scientific discourse but could be made more precise by clearly framing the hypothesis.

The phrase "Central Queensland’s Clarke-Connors Range" is somewhat long and cumbersome. It could be shortened or rephrased for better readability. For instance, "Clarke-Connors Range in Central Queensland" might flow more naturally.

The title doesn’t mention the methodology or approach (e.g., statistical modeling, geographic information system (GIS) analysis), which is often included in technical papers to give a sense of the study's framework

++Simple Summary:

The summary claims that accidents happened "randomly," but this is a broad statement. It would be more scientifically accurate to specify the methodology used to conclude that the accidents were random (e.g., statistical analysis or spatial distribution modeling).

The term "high-quality koala habitat" is vague. A clearer definition or explanation of what constitutes high-quality habitat (e.g., specific tree species, water availability, etc.) would strengthen the scientific message.

The timeframe (October 2014 to November 2023) is accurate, but the summary doesn’t explain how much data was analyzed (e.g., the number of koala collisions). Including basic statistics would make the summary more informative.

The phrase “landscape is relatively uniform” is a technical assumption that might need clearer validation. What measures or indices were used to determine this uniformity?

The recommendation for "more research, including more detailed habitat mapping," is good but vague. Mentioning specific tools or methods (like GIS or remote sensing) would enhance the technical depth of the summary.

The phrase "one of the biggest dangers" could be made more formal and specific. A better alternative might be: "One of the major threats to wild koalas."

The statement "koalas aren’t forced to cross the road at specific points" could be refined for clarity. A more scientific phrasing would be: "Koalas appear to cross the road at various points, rather than at designated or frequent crossing areas."

The use of the word "randomly" might give an impression of complete unpredictability. A phrase like “no clear pattern of collisions was identified” could better reflect the results while avoiding potential misinterpretation.

The repetition of "koalas getting hit by cars" could be avoided by varying the language, such as using "vehicle collisions" or "road mortality" for readability and flow.

Innovation: While the study addresses a well-known problem (vehicle collisions as a threat to wildlife), its focus on "landscape homogeneity" and its relationship to random accident distribution is a novel angle. However, the summary does not fully emphasize what is innovative about this specific approach or finding. Highlighting why this study is different from previous koala road mortality research (e.g., in less uniform or urbanized landscapes) would underscore its contribution.

++Abstract:

The assertion that "koala road strike was randomly distributed" is a bold conclusion, but the abstract doesn’t describe how randomness was tested. Was spatial or statistical analysis (e.g., Poisson distribution, kernel density estimation) used to verify randomness? Providing more insight into the analysis method would strengthen the claim.

The identification of "high-quality koala habitat" as the only predictor is significant, but the abstract should clarify what constitutes "high-quality habitat" to avoid vagueness. Also, were other factors like traffic volume, road speed, or weather conditions tested and excluded?

The potential national significance of this koala population is mentioned, but the abstract lacks context on why this population is unique or important. Adding this would help justify the study's relevance.

The term "candidate predictors" is used without specifying which other predictors were considered and ruled out. Listing them briefly would provide more technical depth and show the rigor of the analysis.

While the concept of landscape homogeneity is interesting, the abstract doesn't explain how this was measured or compared to urbanized areas. Mentioning specific landscape features or metrics would make the analysis more robust.

The abstract uses both past ("were analysed") and present tense ("suggesting"). Maintaining consistency by using either past or present tense throughout would improve readability. Preferably, scientific abstracts should use the past tense to describe findings.

The sentence "koalas are not funneled to particular crossing points by habitat features" could be simplified for clarity. For example, "koalas do not concentrate at specific crossing points due to habitat features" would be clearer and more direct.

The phrase "more detailed habitat mapping" could be streamlined to "detailed habitat mapping" as the word "more" is redundant.

The term "neglected population" is somewhat vague. It would be better to specify in what way the population has been neglected (e.g., under-studied, lacking conservation attention)

++Introduction

While the introduction covers several important issues related to wildlife-vehicle collisions (WVCs), the scope is somewhat unfocused. It transitions from general WVCs to a specific focus on koalas without clearly delineating the importance of each topic. A stronger framing of the introduction would guide the reader from a general understanding of WVCs to the specific conservation issue of koalas, highlighting why koalas require particular attention.

The introduction provides a good overview of existing knowledge but fails to explicitly highlight gaps in the literature. A more effective introduction would point out the lack of comprehensive studies on koala WVCs in the Clarke-Connors Range to justify the need for this study.

The introduction could be more effective by explaining why the koala population in the Clarke-Connors Range is significant in a national or regional context. This would better justify the study’s focus on this population.

The introduction mentions the difficulty of obtaining data on animal casualties but does not explain how these limitations were addressed in this study. Including a brief explanation of the study's improved dataset or methodology would add technical depth.

While the introduction mentions habitat and road design as factors in WVCs, it could provide a clearer overview of which variables will be tested in the study. For example, what specific road characteristics (e.g., speed limits, traffic volume) are being analyzed, and how will habitat quality be quantified?

The reference to "more sophisticated statistical tools" is vague. The introduction would benefit from a brief mention of what kind of statistical methods or models (e.g., spatial regression, GIS-based hotspot analysis) will be used to improve upon previous research.

The introduction could improve its flow by organizing the information into more distinct sections. For instance, starting with a broader discussion on WVCs and narrowing down to koalas and then to the specific case of the Clarke-Connors Range would create a more logical progression.

Phrases like "koalas are an endangered species in Australia, with vehicle strike a key threatening process" are repeated in different forms. These could be consolidated to avoid redundancy and streamline the narrative.

The sentence, “Koalas living in developed areas must cross streets and highways to get to relic habitat pockets” could be more scientifically framed, such as "Koalas residing in urbanized landscapes often traverse roads to access fragmented habitat patches, increasing their susceptibility to WVCs."

The scientific name Phascolarctos cinereus should be italicized to conform to biological nomenclature standards.

Conclusion: Overall, the introduction covers relevant topics but could benefit from more focus, clearer articulation of research gaps, and greater emphasis on the specific contributions of the study. It offers a solid foundation but requires refinement to better justify the need for the research and its novelty.

++Materials and Methods

The reliance on chainage points every 5 km might miss finer spatial patterns or nuances in strike locations that occur between these points. Additionally, the assumption that drivers maintain legal speed limits is a simplification, which may introduce some bias, as actual driving behavior is often variable.

Daytime measurements are used to estimate nighttime visibility, which may introduce significant inaccuracies, as koala strikes often occur during low-light conditions. This could affect the reliability of this predictor.

The koala strike data from wildlife carers and TMR may introduce reporting bias, as not all strikes are equally reported or discovered. Reliance on data from just two sources (responsible for 82% of records) may limit the generalizability of the results.

The text is generally clear, but some terms (e.g., "chainage points") might require brief explanations for readers unfamiliar with specific terminology related to road infrastructure.

There is slight inconsistency in the presentation of certain terms, such as "high-quality koala habitat" vs. "high quality class." It would be clearer to maintain uniform phrasing throughout.

++Results:

Driver Visibility and Road Width: These factors were found to be spatially autocorrelated, but their lack of significance in the final model is underexplored. The analysis would benefit from a more thorough discussion of why these predictors were spatially autocorrelated and how they might still play a role in koala strikes, even if not statistically significant.

Spatial Analysis: The KDE+ and linear K function approaches are appropriate for identifying clustering and distribution patterns. However, the bandwidth (<1 km) used in the KDE+ tool may be too small to detect more widely distributed clusters. Testing the model with different bandwidths or more advanced clustering methods could yield different insights.

++Discussion:

The discussion could more deeply explore why certain predictors (like road width and driver visibility) were insignificant despite initial hypotheses suggesting otherwise. Simply stating that these factors were insignificant without further exploration of their ecological relevance weakens the overall interpretation.

The discussion briefly touches on the coarseness of current habitat mapping but does not sufficiently explore the implications of this limitation. For example, the possibility that fine-scale habitat variables (like individual trees or microhabitats) may play a significant role in koala strikes should be more thoroughly explored.

While the text mentions koala-sensitive infrastructure, it does not provide specific management recommendations based on the study’s findings. More practical suggestions could be provided for policymakers and infrastructure designers to mitigate koala vehicle strikes.

The comparison between SEQ and the PDH study site is an important aspect of the discussion. However, it lacks sufficient detail about how differences in landscape and road conditions might lead to the contrasting patterns observed.

While the discussion provides an overview of the significant results (e.g., habitat quality as a predictor of strikes), it does not adequately explain the absence of hotspots or clustering of koala strikes. The randomness of vehicle strikes across a homogeneous landscape is a key result but needs deeper explanation.

The section is mostly clear, but several long and convoluted sentences could be simplified for better readability. For example, sentences spanning several lines could be broken down to focus on one main idea per sentence.

Certain phrases, such as “high quality koala habitat,” are repeated frequently. Varied phrasing or combining these references into a more concise statement could improve the flow of the discussion.

The paragraphs are generally well-structured, but some are overly long, particularly when multiple ideas are being discussed. For instance, the discussion about the role of landscape heterogeneity could be broken into more digestible sections.

Some of the figures (like Figure 5) are referenced in the discussion but not sufficiently explained in the text. A more detailed explanation of how these figures support the main arguments would improve the presentation.

Limited Novelty: The study's novelty lies more in the specific regional focus and dataset than in the methodological approaches used. The discussion largely confirms existing knowledge about the influence of habitat on koala mortality and does not introduce significantly new concepts or breakthroughs. While the findings add to regional conservation knowledge, the lack of novel insights limits the overall scientific contribution.

++Conclusions

The conclusion summarizes key findings and aligns with the research objectives, particularly regarding the random distribution of koala WVC (wildlife vehicle collisions) and the hypothesis that habitat homogeneity is a key driver. The discussion about the impact of homogeneous landscapes on WVCs is valid and could have wider applicability. However, the conclusion could benefit from more nuanced discussion of other factors influencing WVCs, such as koala behavior and seasonal movements. Additionally, the potential significance of the local koala population to national conservation efforts is mentioned but needs further support, perhaps with references to population trends or comparisons to other regions.

The suggestion to avoid areas of high-quality koala habitat and to implement diversionary infrastructure is appropriate but needs to be more specific. Details about possible mitigation strategies, such as underpasses, overpasses, or roadside barriers, would enhance the conclusions. Additionally, the role of rail corridors could be explored further, given the mention of their similarity to road networks in terms of WVCs.

The writing is generally clear, but there are some awkward phrases. For example:

"It is, perhaps, a population of national significance, but we lack basic knowledge" could be revised for flow.

The use of "WVC" in multiple sentences is redundant and can be simplified. A more fluid sentence structure would help the readability.

Coherence & Structure:The conclusion is generally coherent but could benefit from a clearer structure. For example:

Begin by summarizing the key findings concisely.

Then discuss implications for conservation and management strategies.

Finally, highlight areas for future research.

Comments on the Quality of English Language

++Title:

The phrase "may drive" suggests uncertainty, which is acceptable in scientific discourse but could be made more precise by clearly framing the hypothesis.

The phrase "Central Queensland’s Clarke-Connors Range" is somewhat long and cumbersome. It could be shortened or rephrased for better readability. For instance, "Clarke-Connors Range in Central Queensland" might flow more naturally.

The title doesn’t mention the methodology or approach (e.g., statistical modeling, geographic information system (GIS) analysis), which is often included in technical papers to give a sense of the study's framework

++Simple Summary:

The summary claims that accidents happened "randomly," but this is a broad statement. It would be more scientifically accurate to specify the methodology used to conclude that the accidents were random (e.g., statistical analysis or spatial distribution modeling).

The term "high-quality koala habitat" is vague. A clearer definition or explanation of what constitutes high-quality habitat (e.g., specific tree species, water availability, etc.) would strengthen the scientific message.

The timeframe (October 2014 to November 2023) is accurate, but the summary doesn’t explain how much data was analyzed (e.g., the number of koala collisions). Including basic statistics would make the summary more informative.

The phrase “landscape is relatively uniform” is a technical assumption that might need clearer validation. What measures or indices were used to determine this uniformity?

The recommendation for "more research, including more detailed habitat mapping," is good but vague. Mentioning specific tools or methods (like GIS or remote sensing) would enhance the technical depth of the summary.

The phrase "one of the biggest dangers" could be made more formal and specific. A better alternative might be: "One of the major threats to wild koalas."

The statement "koalas aren’t forced to cross the road at specific points" could be refined for clarity. A more scientific phrasing would be: "Koalas appear to cross the road at various points, rather than at designated or frequent crossing areas."

The use of the word "randomly" might give an impression of complete unpredictability. A phrase like “no clear pattern of collisions was identified” could better reflect the results while avoiding potential misinterpretation.

The repetition of "koalas getting hit by cars" could be avoided by varying the language, such as using "vehicle collisions" or "road mortality" for readability and flow.

Innovation: While the study addresses a well-known problem (vehicle collisions as a threat to wildlife), its focus on "landscape homogeneity" and its relationship to random accident distribution is a novel angle. However, the summary does not fully emphasize what is innovative about this specific approach or finding. Highlighting why this study is different from previous koala road mortality research (e.g., in less uniform or urbanized landscapes) would underscore its contribution.

++Abstract:

The assertion that "koala road strike was randomly distributed" is a bold conclusion, but the abstract doesn’t describe how randomness was tested. Was spatial or statistical analysis (e.g., Poisson distribution, kernel density estimation) used to verify randomness? Providing more insight into the analysis method would strengthen the claim.

The identification of "high-quality koala habitat" as the only predictor is significant, but the abstract should clarify what constitutes "high-quality habitat" to avoid vagueness. Also, were other factors like traffic volume, road speed, or weather conditions tested and excluded?

The potential national significance of this koala population is mentioned, but the abstract lacks context on why this population is unique or important. Adding this would help justify the study's relevance.

The term "candidate predictors" is used without specifying which other predictors were considered and ruled out. Listing them briefly would provide more technical depth and show the rigor of the analysis.

While the concept of landscape homogeneity is interesting, the abstract doesn't explain how this was measured or compared to urbanized areas. Mentioning specific landscape features or metrics would make the analysis more robust.

The abstract uses both past ("were analysed") and present tense ("suggesting"). Maintaining consistency by using either past or present tense throughout would improve readability. Preferably, scientific abstracts should use the past tense to describe findings.

The sentence "koalas are not funneled to particular crossing points by habitat features" could be simplified for clarity. For example, "koalas do not concentrate at specific crossing points due to habitat features" would be clearer and more direct.

The phrase "more detailed habitat mapping" could be streamlined to "detailed habitat mapping" as the word "more" is redundant.

The term "neglected population" is somewhat vague. It would be better to specify in what way the population has been neglected (e.g., under-studied, lacking conservation attention)

++Introduction

While the introduction covers several important issues related to wildlife-vehicle collisions (WVCs), the scope is somewhat unfocused. It transitions from general WVCs to a specific focus on koalas without clearly delineating the importance of each topic. A stronger framing of the introduction would guide the reader from a general understanding of WVCs to the specific conservation issue of koalas, highlighting why koalas require particular attention.

The introduction provides a good overview of existing knowledge but fails to explicitly highlight gaps in the literature. A more effective introduction would point out the lack of comprehensive studies on koala WVCs in the Clarke-Connors Range to justify the need for this study.

The introduction could be more effective by explaining why the koala population in the Clarke-Connors Range is significant in a national or regional context. This would better justify the study’s focus on this population.

The introduction mentions the difficulty of obtaining data on animal casualties but does not explain how these limitations were addressed in this study. Including a brief explanation of the study's improved dataset or methodology would add technical depth.

While the introduction mentions habitat and road design as factors in WVCs, it could provide a clearer overview of which variables will be tested in the study. For example, what specific road characteristics (e.g., speed limits, traffic volume) are being analyzed, and how will habitat quality be quantified?

The reference to "more sophisticated statistical tools" is vague. The introduction would benefit from a brief mention of what kind of statistical methods or models (e.g., spatial regression, GIS-based hotspot analysis) will be used to improve upon previous research.

The introduction could improve its flow by organizing the information into more distinct sections. For instance, starting with a broader discussion on WVCs and narrowing down to koalas and then to the specific case of the Clarke-Connors Range would create a more logical progression.

Phrases like "koalas are an endangered species in Australia, with vehicle strike a key threatening process" are repeated in different forms. These could be consolidated to avoid redundancy and streamline the narrative.

The sentence, “Koalas living in developed areas must cross streets and highways to get to relic habitat pockets” could be more scientifically framed, such as "Koalas residing in urbanized landscapes often traverse roads to access fragmented habitat patches, increasing their susceptibility to WVCs."

The scientific name Phascolarctos cinereus should be italicized to conform to biological nomenclature standards.

Conclusion: Overall, the introduction covers relevant topics but could benefit from more focus, clearer articulation of research gaps, and greater emphasis on the specific contributions of the study. It offers a solid foundation but requires refinement to better justify the need for the research and its novelty.

++Materials and Methods

The reliance on chainage points every 5 km might miss finer spatial patterns or nuances in strike locations that occur between these points. Additionally, the assumption that drivers maintain legal speed limits is a simplification, which may introduce some bias, as actual driving behavior is often variable.

Daytime measurements are used to estimate nighttime visibility, which may introduce significant inaccuracies, as koala strikes often occur during low-light conditions. This could affect the reliability of this predictor.

The koala strike data from wildlife carers and TMR may introduce reporting bias, as not all strikes are equally reported or discovered. Reliance on data from just two sources (responsible for 82% of records) may limit the generalizability of the results.

The text is generally clear, but some terms (e.g., "chainage points") might require brief explanations for readers unfamiliar with specific terminology related to road infrastructure.

There is slight inconsistency in the presentation of certain terms, such as "high-quality koala habitat" vs. "high quality class." It would be clearer to maintain uniform phrasing throughout.

++Results:

Driver Visibility and Road Width: These factors were found to be spatially autocorrelated, but their lack of significance in the final model is underexplored. The analysis would benefit from a more thorough discussion of why these predictors were spatially autocorrelated and how they might still play a role in koala strikes, even if not statistically significant.

Spatial Analysis: The KDE+ and linear K function approaches are appropriate for identifying clustering and distribution patterns. However, the bandwidth (<1 km) used in the KDE+ tool may be too small to detect more widely distributed clusters. Testing the model with different bandwidths or more advanced clustering methods could yield different insights.

++Discussion:

The discussion could more deeply explore why certain predictors (like road width and driver visibility) were insignificant despite initial hypotheses suggesting otherwise. Simply stating that these factors were insignificant without further exploration of their ecological relevance weakens the overall interpretation.

The discussion briefly touches on the coarseness of current habitat mapping but does not sufficiently explore the implications of this limitation. For example, the possibility that fine-scale habitat variables (like individual trees or microhabitats) may play a significant role in koala strikes should be more thoroughly explored.

While the text mentions koala-sensitive infrastructure, it does not provide specific management recommendations based on the study’s findings. More practical suggestions could be provided for policymakers and infrastructure designers to mitigate koala vehicle strikes.

The comparison between SEQ and the PDH study site is an important aspect of the discussion. However, it lacks sufficient detail about how differences in landscape and road conditions might lead to the contrasting patterns observed.

While the discussion provides an overview of the significant results (e.g., habitat quality as a predictor of strikes), it does not adequately explain the absence of hotspots or clustering of koala strikes. The randomness of vehicle strikes across a homogeneous landscape is a key result but needs deeper explanation.

The section is mostly clear, but several long and convoluted sentences could be simplified for better readability. For example, sentences spanning several lines could be broken down to focus on one main idea per sentence.

Certain phrases, such as “high quality koala habitat,” are repeated frequently. Varied phrasing or combining these references into a more concise statement could improve the flow of the discussion.

The paragraphs are generally well-structured, but some are overly long, particularly when multiple ideas are being discussed. For instance, the discussion about the role of landscape heterogeneity could be broken into more digestible sections.

Some of the figures (like Figure 5) are referenced in the discussion but not sufficiently explained in the text. A more detailed explanation of how these figures support the main arguments would improve the presentation.

Limited Novelty: The study's novelty lies more in the specific regional focus and dataset than in the methodological approaches used. The discussion largely confirms existing knowledge about the influence of habitat on koala mortality and does not introduce significantly new concepts or breakthroughs. While the findings add to regional conservation knowledge, the lack of novel insights limits the overall scientific contribution.

++Conclusions

The conclusion summarizes key findings and aligns with the research objectives, particularly regarding the random distribution of koala WVC (wildlife vehicle collisions) and the hypothesis that habitat homogeneity is a key driver. The discussion about the impact of homogeneous landscapes on WVCs is valid and could have wider applicability. However, the conclusion could benefit from more nuanced discussion of other factors influencing WVCs, such as koala behavior and seasonal movements. Additionally, the potential significance of the local koala population to national conservation efforts is mentioned but needs further support, perhaps with references to population trends or comparisons to other regions.

The suggestion to avoid areas of high-quality koala habitat and to implement diversionary infrastructure is appropriate but needs to be more specific. Details about possible mitigation strategies, such as underpasses, overpasses, or roadside barriers, would enhance the conclusions. Additionally, the role of rail corridors could be explored further, given the mention of their similarity to road networks in terms of WVCs.

The writing is generally clear, but there are some awkward phrases. For example:

"It is, perhaps, a population of national significance, but we lack basic knowledge" could be revised for flow.

The use of "WVC" in multiple sentences is redundant and can be simplified. A more fluid sentence structure would help the readability.

Coherence & Structure:The conclusion is generally coherent but could benefit from a clearer structure. For example:

Begin by summarizing the key findings concisely.

Then discuss implications for conservation and management strategies.

Finally, highlight areas for future research.

Reviewer 3 Report

Comments and Suggestions for Authors

The study is interesting, well detailed and methodologically sound. However, given the objective of supporting koala conservation at the study site, I think the authors could expand their recommendations for risk mitigation measures for koala coalitions.

Introduction

Line 71. Change to italics Phascolarctos cinereus

Line 82. Rewrite the sentence containing reference 28 so that it does not begin with the number in brackets.

Line 86. Rewrite the sentence containing reference 29 so that it does not begin with the number in brackets.

Line 96. Rewrite the sentence containing reference 25 so that it does not start with the number in brackets.

Line 99. There is a double space at the beginning of the sentence, delete it.

- Missing hypothesis

Materials and methods

Line 123. Delete the r in hr, it should be 100km/h.

Line 137. What is meant here: Two people were responsible for 82% of the recorded koalas. Does this mean two keepers? clarify.

Line 147. Include R's version.

Line 165. Include the appendix as supplementary material.

Line 203. There is an extra space at the beginning of the second sentence.

Line 206. There is an extra space at the start of the sentence.

Line 208. Delete the r in hr.

Line 209. Delete the r in hr.

Line 239. Insert the version of R.

Line 240. Delete 3. Results from the end of the sentence.

The results

Line 251. Insert the interval in months of the koala breeding season.

Lines 251-252. Improve the map in Figure 1. The text is not clear. Also check that the scale is correct, it seems larger than it should be.

Lines 256-257. Figure 2. Delete the outermost box in the figure.

Line 259. Do not start a sentence with a number. Change 215 to two hundred and fifteen.

Lines 273-274. Improve the map.

Line 286. There is an extra space at the beginning of the sentence.

Line 289. There is an extra space at the start of the sentence.

Lines 306-307. Improve the map.

Discussions

Line 318. In the discussion you mention a hypothesis, but it will be necessary to state this hypothesis from the introduction as part of the statement of the research problem.

Lines 319-322. It is mentioned that: However, we were unable to detect any statistically significant hotspots across our study area. This may reflect a random distribution of road strikes (except at very large thresholds), with koalas equally likely to be hit by a vehicle at any point along the highway. However, Figure 2 states that "... koala vehicle strikes were associated with greater areas of high quality koala habitat". Please clarify this wording to avoid confusion.

Lines 341-345. So the authors consider the landscape at their study site to be homogeneous? It is not clear how to interpret this part: ... Our study extends this by showing how the distribution of mortality events can also be influenced by a lack of heterogeneity in the landscape. That is, variation in habitat quality and the presence or absence of habitat along a road corridor drives variation in the distribution of vehicle strikes for the species under study.

Line 349. Reword so that the sentence does not begin with the reference number.

Line 350. Reword so that the sentence does not start with the reference number.

Lines 378-381. This is confusing again because it suggests that high quality habitat is an indicator of koala coalition, and even more so because it suggests that high quality habitat near roads may be more of a problem for koalas.

Line 383. Change the wording to try not to start the sentence with the reference number.

Line 385. Rewrite to try not to start the sentence with the reference number.

Lines 412-413. This is correct, although I suggest that the description of what happens at some of the sites, particularly those marked with red circles (46-51 koala vehicle strikes), be expanded a little more.

I think that the final part of the discussion should include some recommendations for risk mitigation, as mentioned in lines 457-465 and suggested by Pagany (2020), and expand and detail the information specifically for the study site.

Conclusions.

Lines 457-465. Consider whether this information can be part of the recommendations at the end of the discussion.

Line 464. It is better to specify what the inclusion of AI refers to.

Round 2

Reviewer 2 Report

Comments and Suggestions for Authors

No comments.